# Detection and characterization of birch pollen in the atmosphere using multi-wavelength Raman polarization lidar and Hirst-type pollen sampler in Finland

Stephanie Bohlmann[1], Xiaoxia Shang[1], Elina Giannakaki[1,2], Maria Filioglou[1], Annika Saarto[3], Sami Romakkaniemi[1] and Mika Komppula[1]

[1]Finnish Meteorological Institute, P.O. Box 1627, 70211 Kuopio, Finland
[2]Department of Environmental Physics and Meteorology, University of Athens, 15784 Athens, Greece
[3]Biodiversity Unit, University of Turku, 20014 Turku, Finland

*Correspondence to*: Stephanie Bohlmann (stephanie.bohlmann@fmi.fi)

**Abstract.** We present the results of birch pollen characterization using lidar and in-situ measurements based on a 11-day pollination period from 5 to 15 May 2016 at the European Aerosol Research Lidar Network (EARLINET) station in Vehmasmäki (Kuopio, 62°44′N, 27°33′E), Finland. The ground-based multi-wavelength Raman polarization lidar Polly$^{XT}$ performed continuous measurements at this rural forest site and has been combined with a Hirst-type volumetric air sampler which measured the pollen type and concentration on roof level (4 m). The period was separated into two parts due to different atmospheric conditions and detected pollen types. During the first period, high concentrations of birch pollen were measured with a maximum two-hour average pollen concentration of 3700 grains/m³. Other pollen types represented less than 3% of the total pollen count. In observed pollen layers, the mean particle depolarization ratio at 532 nm was $10 \pm 6\%$ during the intense birch pollination period. Mean lidar ratios were found to be $45 \pm 7$ and $55 \pm 16$ sr at 355 and 532 nm, respectively. During the second period, birch pollen was still dominant but a significant contribution of spruce pollen was observed as well. Spruce pollen grains are highly non-spherical, leading to a larger mean depolarization ratio of $26 \pm 7\%$ of the birch-spruce pollen mixture. Furthermore, higher lidar ratios were observed during this period with a mean value of $60 \pm 3$ and $62 \pm 10$ sr at 355 and 532 nm, respectively. The presented study shows the potential of the particle depolarization ratio to track pollen grains in the atmosphere.

## 1 Introduction

Atmospheric pollen is a well-known health threat as it can irritate the respiratory system and cause asthmatic symptoms (Bousquet et al., 2008). The number of people suffering from pollen triggered diseases is rising (Schmidt, 2016) and the prevalence of pollen allergies is likely to further increase due to climate change as the pollination season becomes longer and the pollen production increases (Lake et al., 2018). In addition to the well-known allergenic impacts, pollen also affect the climate (IPCC, 2013; WHO, 2003). Steiner et al. (2015) suggested that fragments of pollen act as cloud condensation nuclei

(CCN) and therefore influence cloud optical properties. Pollen can furthermore change ice cloud formation processes by acting as ice nuclei (IN) (von Blohn et al., 2005; Diehl et al., 2001, 2002).

Worldwide, 879 active stations continuously monitor pollen type and concentration near ground level in 2016 (Buters et al., 2018). The majority of these stations operate Hirst-type volumetric air sampler. These traditional pollen traps are operated manually, which requires human resources and is time consuming. In the recent years, novel techniques have been developed to enable automated pollen monitoring and reduce workload. Those techniques use, for example, automated image recognition (Oteros et al., 2015) or fluorescence spectra (Crouzy et al., 2016; Richardson et al., 2019; Saito et al., 2018) to identify pollen types and could enable a systematic pollen monitoring on ground level in near real-time. Systematic information on the vertical distribution, however, is missing. Models are using phenological and meteorological data to forecast concentration and dispersion of pollen in the atmosphere. But observations, especially in the vertical direction, to evaluate the model results are rare or missing. Light detection and ranging (lidar) is an effective method to investigate the vertical distribution of aerosols, as it enables measurements with high vertical and temporal resolution under ambient conditions. Recently, the signature of pollen grains has been observed in lidar measurements (Noh et al., 2013a, 2013b; Sassen, 2008; Sicard et al., 2016). It has been revealed that non-spherical pollen generate strong laser depolarization and thus the information on particle shape can be retrieved. For example, Sassen (2008) measured a linear depolarization ratio at 694 nm up to 30% for paper birch in Alaska. In controlled laboratory experiments, Cao et al. (2010) measured the linear depolarization ratio of different pollen types and determined a linear depolarization ratio at 532 nm for paper birch of 33%. In the absence of other depolarizing particles, the depolarization ratio could therefore be used to track pollen grains. Lidar-derived depolarization ratio profiles thus can provide information about the vertical distribution of pollen, which could result in an improvement of the model input parameters and serve as validation for model results.

Within our study, we aim to improve and deepen the knowledge of optical properties of pollen in the atmosphere by using a multi-wavelength Raman polarization lidar. Finland provides suitable conditions for the observation of pollen as 78% of Finland's total area is forest land and sources of other highly depolarizing particles like dust are absent. Hence the contamination with other aerosols is considered to be small. During a four months measurement campaign in 2016, the multi-wavelength Raman polarization lidar Polly[XT] (Engelmann et al., 2016) performed continuous measurements at the rural forest station in Vehmasmäki (Kuopio) which is part of the European Aerosol Research Lidar Network (EARLINET). Simultaneously, a Hirst-type volumetric air sampler was operated to obtain pollen type and concentration at roof level. Twenty-one different pollen types were detected from May to August 2016. In this study, we focus on the description of birch pollen and the mixture of birch and spruce pollen as birch is one of the most allergenic pollen types and the most allergenic tree pollen in northern, central and eastern Europe (D'Amato et al., 2007).

## 2 Measurement site and instrumentation

Our measurement campaign took place from the beginning of May to the end of August 2016 at Vehmasmäki, Finland (62°44′ N, 27°33′E, 190 m asl); a rural forest site 18 km from the city center of Kuopio, Eastern Finland. The measurement site was equipped with a multi-wavelength Raman polarization lidar Polly$^{XT}$ (Sect.2.1) and a Hirst-type pollen sampler (Sect.2.2). With this set-up it is possible to combine vertical profiles of the aerosol properties above the site and the pollen concentration on ground. Due to the location of the site, far from major aerosol sources like dust or anthropogenic aerosol, mainly surrounded by forest, the atmosphere is relatively clean and suitable for pollen studies.

### 2.1 Lidar: Polly$^{XT}$

Lidar measurements were conducted with the multi-wavelength Raman polarization lidar Polly$^{XT}$ (Althausen et al., 2009; Baars et al., 2016; Engelmann et al., 2016). Polly$^{XT}$ has three emission wavelengths (355, 532 and 1064 nm) and seven detection channels. In addition to the three emitted wavelengths, the backscattered signals at the in-elastic Raman-shifted wavelengths (387, 407 and 607 nm) and the cross-polarized signal at 532 nm are detected. During nighttime, extinction and backscatter coefficient profiles at 355 and 532 nm can be determined independently using the Raman method (Ansmann et al., 1992). During daytime, the Klett-Fernald method (Fernald, 1984; Klett, 1981) is applied using the elastic signals due to the low signal-to-noise ratio at the Raman channels. The signal at the Raman-shifted wavelength (407 nm) is used to determine water vapor mixing ratio profiles during dark hours (Filioglou et al., 2017). The simultaneous measurement of the cross-polarized and total backscattered light at 532 nm enables the determination of the linear particle depolarization ratio (PDR, Freudenthaler et al. (2009)), which allows the characterization of particle shape (Sassen, 2005). The measurement of multiple wavelengths allows the retrieval of Ångström exponents (Å), which are related to the particle size. The ratio of extinction to backscatter coefficient is called lidar ratio (LR). It is considered an important criterion for particle characterization, as it depends on single scattering albedo and backscatter phase function and therefore on the size distribution and the chemical composition of the aerosol particle. The LR is therefore considered to be aerosol type dependent.

The operated lidar system has an initial spatial resolution of 30 m and a temporal resolution of 30 s. Due to the bi-axial set-up of emission and detection units, the height of complete overlap between the laser and the receiver field-of-view is reached at about 800-900 m (Engelmann et al., 2016). An overlap correction can be applied on the basis of a simple technique proposed by Wandinger and Ansmann (2002), which allows to extend profiles down to around 500 m. In this study, the lower limit of reliable profiles of vertically smoothed and temporally averaged optical properties is at around 800 m. Uncertainties in nighttime lidar-products are mainly determined by signal noise and the correction of Rayleigh scattering. The overall relative errors of the lidar-derived optical properties retrieved with the Raman method are in the range of 5–10% for backscatter coefficients and depolarization ratios and 10–20% for extinction coefficients (Ansmann et al., 1992; Baars et al., 2012). These uncertainties propagate to the retrieved Ångström exponents and LRs.

Further details on the set-up, principle and error propagation of Polly[XT] can be found in Althausen et al. (2009) and Engelmann et al. (2016). Near-real-time measurements and Polly[XT] data can be accessed at the PollyNET website (http://polly.tropos.de/).

## 2.2 Pollen collector: Hirst-type volumetric air sampler

A Hirst-type volumetric air sampler located next to the lidar, 4 m above ground, monitored the pollen concentration and type. This type of spore sampler enables continuous 7-day collection of pollen grains with 2 hour time resolution. The sampling principle is based on the design described by Hirst (1952). With a flow rate of 10 L/min air is drawn into the sampling device through a 14 mm × 2 mm orifice. A large wind vane on a rotatable sampler head makes the sampler sensitive to changes in wind direction and ensures that the orifice is always oriented towards the wind. Particles impact on an

adhesive-coated plastic tape beneath the orifice. For this study, the tape, fixed on a rotating drum, was changed every seven days and the pollen grains impacted on the tape were further analyzed under the microscope. The pollen type was determined using characteristic features of the examined pollen grains. By converting the counted spores on the sample tape surface in relation to the inlet air flow, the pollen concentration was obtained.

## 3 Methodology

Figure 1 shows the temporal variation of the pollen concentration (a), the range corrected signal at 1064 nm (b) and the volume depolarization ratio at 532 nm (c) during the period 5 – 15 May 2016. This period represents the main birch pollination season of 2016 as 83% of the annual birch pollen have been collected during this time. A relatively large aerosol load was observed within the planetary boundary layer up to ~ 3.5 km. As shown in Fig. 1c, the volume depolarization ratio ranges between 4-10% suggesting the presence of non-spherical particles. A detailed examination of the air masses arriving

during this period along with modelled dust load using the BSC-DREAM8b model (Basart et al., 2012) confirms the absence of dust in middle and northern Europe. Additionally, MODIS data (MODIS, 2019) were synergistically used to exclude smoke aerosol layers from biomass burning. Hence, the highly depolarizing aerosol layers were likely attributed to pollen, keeping in mind that some contamination with local anthropogenic aerosol is always possible.

Ground-level pollen concentration values presented in Fig.1 (a) were used to verify the strong pollination event in the

beginning of May which provides 50% of the annual birch pollen concentration. The event started in the evening (17:00 UTC) of 5 May and lasted until 9 May noon (hereafter called period 1). During period 1, the two-hour average pollen concentration exceed 1000 grains/m³ in 53% of the time. The majority of pollen type identified was birch (97%) with a very small contribution of willow (2%) and other pollen types (1%). From 12 to 15 May (period 2), the mean pollen concentration was significantly lower. Only 8% of the time, the total pollen concentration was higher than 1000 grains/m$^3$. In addition to

birch (82%), spruce pollen (14%) and other pollen types (4%) were detected. This variation can be explained by the different meteorological conditions during the two periods. A different predominant wind direction during the two periods was

observed, which probable cause the different mixture of pollen types. The most frequent wind direction in period 1 was northwest, whereas in period 2 the air masses were mainly advected from southeast. When comparing the diurnal cycle of temperature and relative humidity measured at ground level, we found higher temperature values and lower relative humidity during period 1 compared with period 2. Temperature and pollen concentration have been shown to be positive correlated,

whereas pollen concentration and relative humidity show a negative correlation (Bartková-Ščevková, 2003). The different pollen concentration could therefore be partly explained by variations of temperature and humidity.

The ground-near aerosol layers are assumed to contain the highest concentration of local pollen and are defined as pollen layers in this study. The gradient method was applied to determine the bottom and top layer heights of the pollen layers (Bösenberg and Matthias, 2003; Flamant et al., 1997; Mattis et al., 2008). The local maximum in the first derivative of the

1064 nm backscatter coefficient was considered to be the bottom of the layer. The local minimum was considered to be the layer top. To verify the determined layers, the layer boundaries identified by the gradient method were compared with the bottom and top heights of coherent structures in the height-time illustration of the range-corrected signal (Giannakaki et al., 2015). The layer identification was based on the assumption that the optical properties should be relatively homogeneous, which means within one layer, the variability of the optical properties should be lower than the statistical uncertainty of the

individual data points. Two layers with a vertical distance less than 100 m apart from each other were combined to one layer. All layers detected during the 11 days period are shown in Fig. 2. Black, magenta, blue and yellow bars show the first, second, third and fourth layer, respectively. Triangles mark the part of the layer which was used for calculations of the mean optical properties of the layer. The lower limit for reliable profiles during our measurement period was at around 800 m. Since the closest layer to the ground is assumed to contain the highest pollen concentration and share, we only consider the

lowest layers (black) in the following analysis.

## 4 Results & Discussion

### 4.1 Case studies

We present two case studies representative for different pollen mixtures: in the first case study only birch pollen have been detected by the Hirst-type sampler. In the second case study, spruce pollen was detected in addition to birch. In the choice of

case studies, backward trajectories have been considered to select cases with minimal contamination with of other aerosol. Furthermore nighttime Raman measurements were chosen to present all lidar-derived parameters including the retrieved LR profile. Figure 3 shows, from left to right, the particle backscatter coefficient at 355 (blue), 532 (green) and 1064 nm (red); the particle extinction coefficient at 355 (dashed blue) and 532nm (dashed green); the LR at 355 (blue) and 532 nm (green); the PDR at 532 nm (light green); the Ångström exponents calculated both from the backscatter coefficient at 355-532nm

(blue) and 532-1064 nm (red) and from extinction coefficients at 355-532 nm (black); and the relative humidity from lidar-derived water vapor profiles (black) and temperature profiles from radiosonde launched at 18 UTC (orange). Lidar-derived

optical properties were vertically smoothed using a sliding average of 25 bins (750 m). Four-day backward trajectories ending at the height of the layers and the middle of the time period are shown as well.

The first case study was selected during the intense birch pollination event (period 1). On 6 May 2016 between 23:00 and 01:00 UTC only birch pollen was detected. Using the layer definition methodology (Sect. 3), three layers were determined and the two lowest ones have been combined to one pollen layer for this analysis since their distance is less than 100 m.

Four-day HYSPLIT (Hybrid Single Particle Lagrangian Integrated Trajectory) backward trajectories (Stein et al., 2015) ending at 450 m and 1.1 km on 7 May 0:00 UTC show that the air masses are advected from western directions and have travelled over the British Isles, the North Sea and southern Sweden. The contamination with depolarizing aerosol like dust is therefore considered to be negligible, however the mixture with other anthropogenic aerosol cannot be ruled out. The presumed birch pollen layer was observed up to 1.2 km. Extinction coefficient at 355 nm is about $22 \pm 2$ Mm$^{-1}$ and $13 \pm 1$ Mm$^{-1}$ at 532 nm. The mean LR for the observed layer is $49 \pm 4$ sr and $70 \pm 7$ sr at 355 and 532 nm, respectively. Mean backscatter and extinction-related Ångström exponents at 355-532 nm are $2.1 \pm 0.04$ and $1.1 \pm 0.5$, respectively. The backscatter-related Ångström exponent between 532 and 1064 nm is $0.9 \pm 0.1$. The mean PDR at 532 nm within the layer is $14 \pm 1\%$. Note that it can be even higher close to the ground, below the height of complete overlap. The PDR decreases with increasing height, while the LR is constant. Thus the measured LR may not be a good indicator for characterizing the observed birch pollen in these cases as the contribution of pollen is assumed to decrease with increasing distance to the pollen source.

During our second case study on 15 May 2016 between 19:00 and 21:00 UTC, spruce pollen have been measured simultaneously to the birch pollen. Profiles and backward trajectories are shown in the lower panel of Fig.3. The pollen layer reaches up to 1.7 km.

The extinction coefficients at 355 and 532 nm are higher than in the previous case, accounting to $61 \pm 5$ Mm$^{-1}$ at 355 nm and $44 \pm 6$ Mm$^{-1}$ at 532 nm. The mean LR is $55 \pm 6$ sr at 355 nm and $51 \pm 9$ sr at 532 nm. The backscatter and extinction-related Ångström exponents at 355-532 nm are lower than in the first case with values of 0.5 to 0.7 and 0.1 to 1.7, respectively. The backscatter-related Ångström exponent at 532-1064 nm is 0.8 to 1.0.

The PDR at 532 nm is on average $19 \pm 2\%$ and 24% at its maximum and is clearly higher than in the case when only birch pollen were observed (period 1). The air masses arriving in the height of the layers on 15 May 20 UTC have been advected from Russia and remained close to the ground only the last 12 hours before reaching the site. The contamination with depolarizing dust can therefore be neglected. An explanation for the higher depolarization of the backscattered light is the non-spherical shape of the spruce pollen grains which have been detected in addition to birch pollen.

Figure 4 shows micrographs of birch and spruce pollen grains. Birch pollen (left picture) have a diameter around 20-30 μm, are almost spherical and possess three pores on the edge of the grain. Spruce pollen grains, on the other hand, possess two air bladders and are clearly non-spherical. Furthermore those pollen grains are significantly larger, with their longest axis diameter ranging between 90 - 110 μm (including air bladders).

Pollen are low density particles, which makes them more sensitive to air currents, reduces the settling velocity and allows them to be lifted by turbulent air flows. Birch pollen, for example, have a gravitational settling velocity of around 1 cm s$^{-1}$ (Sofiev et al., 2006). This settling velocity is similar to anthropogenic aerosol smaller than 10 μm (PM10) although birch pollen grains are more than twice the size. The air bladders on the bigger spruce pollen grains increase the surface area of the grain without adding much mass, and therefore decrease the settling velocity. Hence, even those big pollen grains can be lifted up to several kilometers and be dispersed by wind over thousands of kilometers as have been shown by several studies on the long distance transport of pollen (Rousseau et al., 2008; Skjøth et al., 2007; Szczepanek et al., 2017).

## 4.2 Lidar derived optical parameters

All pollen layers between 5 and 15 May have been identified and analyzed to determine the relationship between pollen type and the lidar-derived optical properties of the aerosol layer. Figure 5a shows the LR at 532 nm against the PDR at the same wavelength for all Raman measurements during nighttime. Measurements during the first intense birch pollination period are shown in green color, while measurements during the second period, when spruce pollen was detected simultaneous to birch pollen, are shown in black. The size of the dots represents the measured pollen concentration by the Hirst-type sampler at ground level. The standard deviation is shown by the error bars. Lidar ratio values range from 31 to 74 sr. This wide range of LRs suggest that the LR alone is not a suitable parameter for the characterization of pollen as other aerosol types also show characteristic values in this range. However, the mean PDR within the pollen layers during the first period is $10 \pm 6\%$, which is significantly higher than that of anthropogenic pollution. In the absence of other depolarizing aerosol, e.g. dust, pollen is therefore likely the dominant aerosol causing this depolarization. Depolarization ratios higher than 15% are only observed during the second period (12-15 May) in which spruce pollen were present. The mean LR and PDR at 532 nm during this period are $62 \pm 10$ sr and $21 \pm 3\%$, respectively. The significantly higher PDR is caused by the non-spherical shape of spruce pollen in those layers. In Fig. 5b, the backscatter-related Ångström exponent between 532 and 1064 nm is shown against the PDR at 532 nm for all measurements between 5 and 15 May, including Klett retrievals for daytime measurements (dotted markers). A clear tendency to smaller Ångström exponents with increasing depolarization ratios in both periods can be seen. This correlation indicates that the influence of non-spherical particles on the backscattered signal increases with decreasing Ångström exponent, i.e. bigger particles. The Ångström exponent in the second period is around 0.8, whereas it is around 1 in the first period, demonstrating the effect of the larger spruce pollen (~90-110 μm), even with a small contribution (~14%) to the total pollen number concentration. But considering the different volume of birch and spruce pollen grains, the contribution of spruce to the total volume concentration exceeds 75% in the second period, which may explain the large effect of spruce pollen on the measured optical properties even with a small number concentration.

However, the effect of the background particles has to be considered. Lidar measurements during the winter months of 2015 and 2016 and during pollen-free periods in spring and summer 2016 have been analyzed to determine the effect of background aerosol at our measurement site. During winter time the absence of pollen can be ensured, but there is a

possibility that pollen also have been present in the atmosphere during spring and summer when no pollen were detected by the Hirst-type sampler on ground. Nevertheless, values of mean PDR at 532 nm are below 4% during all analyzed periods with no observed pollen. Since the PDR during the pollination period is significantly higher than the PDR of the background aerosol, the depolarization ratio can be used as an indicator for detecting the presence of pollen. The Ångström exponent on

the other hand can be also related to the amount and type of background aerosol and is therefore less representative here. Earlier studies show that the relative humidity can affect size and shape of the pollen grains and therefore lead to different optical properties (Franchi et al., 1984; Griffiths et al., 2012; Katifori et al., 2010). When pollen grains dehydrate, the pollen wall can fold onto itself to prevent further dehydration, this phenomena is known as harmomegathy (Katifori et al., 2010). The shape of the pollen grain changes, which could lead to significantly higher depolarization of the backscattered light. At

humid conditions, pollen grains swell by taking up water internally and after reaching a relative humidity over 89% external wetting of the pollen surface can occur (Griffiths et al., 2012). To check whether the ambient relative humidity affects our measurements, the Ångström exponent (532-1064 nm) and the PDR at 532 nm are presented against the relative humidity in Fig. 6. In the selected measurement period, the relative humidity ranged between 40-65%. In this humidity range, Ångström exponent (Fig. 6a) and depolarization ratio (Fig. 6b) do not show any correlation with the relative humidity. Thus, our

measurements were not affected by extreme humidity events and represent values for pollen under ambient atmospheric conditions in the spring season in Finland. However, lidar measurements of relative humidity profiles are only available during nighttime. The relative humidity in the observed pollen layers during daytime could be smaller. This could result in occasional folding of the pollen grains and higher depolarization ratios. This hypothesis could also explain the higher depolarization ratios of about 25% of few Klett measurements of birch pollen during the first period.

Table 1 summarizes the mean intensive properties together with the associated standard derivation (STD), range and median in the first (birch) period of our campaign. The contribution of other pollen types in this period was small. Those values, therefore, can be considered to be characteristic for birch pollen dominated aerosol conditions. Table 2 shows the same properties for the spruce contaminated period. Lidar ratio and PDR are higher when spruce is detected simultaneously with birch. The PDR values for birch pollen are considerably lower than previously determined in lidar studies. A linear

depolarization ratio up to 30% at 694 nm was detected by Sassen (2008) for paper birch in Alaska. And under controlled laboratory environment, Cao et al. (2010) measured a linear depolarization ratio at 532 nm of 33% for dried paper birch pollen. We assume that those high depolarization values can be caused by dry birch pollen grains, which fold and change their shape when dehydrating. Under ambient conditions the pollen grains are more spherical and therefore less depolarizing. Also the orientation of the pollen grains in the atmosphere has to be considered. Pollen with air bladders, e.g. spruce pollen,

are known to align with their air bladders upwards when drifting in the air (Schwendemann et al., 2007) and also an orientation of the almost spherical birch pollen grains was observed (Sassen, 2011; Tränkle and Mielke, 1994). This could cause differences in the measured optical properties if the orientation of the particles in laboratory experiments is not considered and the irregularly shaped particles are observed from different angles.

## 5 Conclusion

Particle depolarization ratios of about 10% have been observed during a birch pollination event in Vehmasmäki, Finland. When more non-spherical pollen, e.g. spruce, are present, the particle depolarization ratio can be as high as 26%. Those depolarization ratios are similar to dust and biomass burning aerosol mixtures (Tesche et al., 2011) or dust mixtures with

marine aerosol (Groß et al., 2011), thus pollen could easily be misclassified as dusty mixtures. The mean LRs show a wide range of values depending on the mixing of different pollen types in the atmosphere. The mean LR at 355 nm vary between $46 \pm 8$ sr (first period) and $60 \pm 3$ sr (second period) and at 532 nm between $52 \pm 12$ sr (first period) and $62 \pm 10$ sr (second period). Those LRs are characteristic for dust or dust-smoke mixtures (Tesche et al., 2011), which complicates the characterization of pollen using the LR. Also the backscatter-related Ångström exponent at 532-1064 nm, which is around

1.0 for the intense birch pollination period and around 0.8 for the spruce contaminated period, is similar to characteristic values for smoke and dust-smoke mixtures, respectively. Thus, in order to distinguish between pollen and other aerosol types, all three parameters and backward trajectories as well as possible dust and biomass-burning aerosol sources have to be considered.

The presented data show the potential of lidar measurements to detect pollen in the atmosphere. Nevertheless, there are

challenges which need to be addressed in order to improve the characterization of optical properties of airborne pollen. First, the minimum height of the usable lidar signal needs to be as low as possible. By operating a lidar system with a low full-overlap height or additional near-field channels, the coverage of lower heights can be significantly improved. Second, the contribution of other aerosol types like anthropogenic pollution has to be determined. Therefore, more multi-wavelength lidar studies with depolarization characterization on atmospheric pollen are necessary.

**Data availability**

Lidar data is available upon request from the authors and data quicklooks are available on PollyNET website (http://polly.tropos.de/). Trajectories are calculated with the NOAA (National Oceanic and Atmospheric Administration) HYSPLIT (HYbrid Single-Particle Lagrangian Integrated Trajectory) model (https://ready.arl.noaa.gov/HYSPLIT.php, accessed: 30/04/2019). Fire data is available at the NASA Worldview application (https://worldview.earthdata.nasa.gov,

accessed: 30/04/2019). BSC-DREAM8b model simulations are operated by the Barcelona Supercomputing Center and are available at https://ess.bsc.es/bsc-dust-daily-forecast/ (accessed: 30/04/2019).

**Author contribution**

SB, XS and MF performed the lidar data analysis. AS analyzed the pollen samples. MK and EG initiated the measurement campaign. All authors contributed to the scientific discussion and the manuscript preparation.

**Competing interests**

The authors declare that they have no conflict of interest.

**Acknowledgement**

This project was supported by the Academy of Finland (project no. 310312). We acknowledge the use of data and imagery
of BSC-DREAM8b simulations performed by the Barcelona Supercomputing Center and from the NASA Worldview
application (https://worldview.earthdata.nasa.gov), part of the NASA Earth Observing System Data and Information System
(EOSDIS). We thank the NOAA Air Resources Laboratory (ARL) for the provision of the HYSPLIT transport and
dispersion model used in this publication.

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

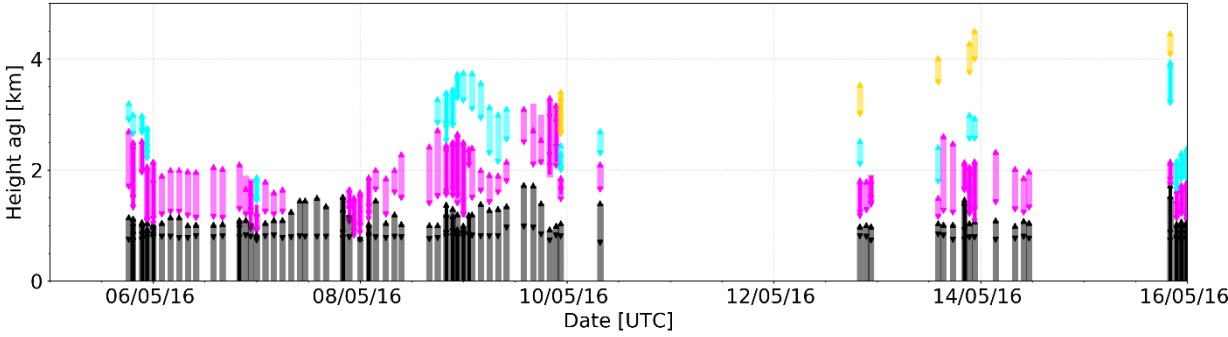

**Figure 1: Overview of the 11 days birch pollen period. (A) Pollen concentration obtained from Hirst-type pollen sampler, (B) Range-corrected signal at 1064 nm and (C) Volume depolarization ratio at 532 nm. Dashed vertical lines mark the period of the case studies.**

**Figure 2: Layer definition during the measurement period 5-15 May. Definition of layers explained in Sect. 3. Colors mark the upper layers, which were not used for further analysis. Triangles mark lower and upper limit of the area used for calculation.**

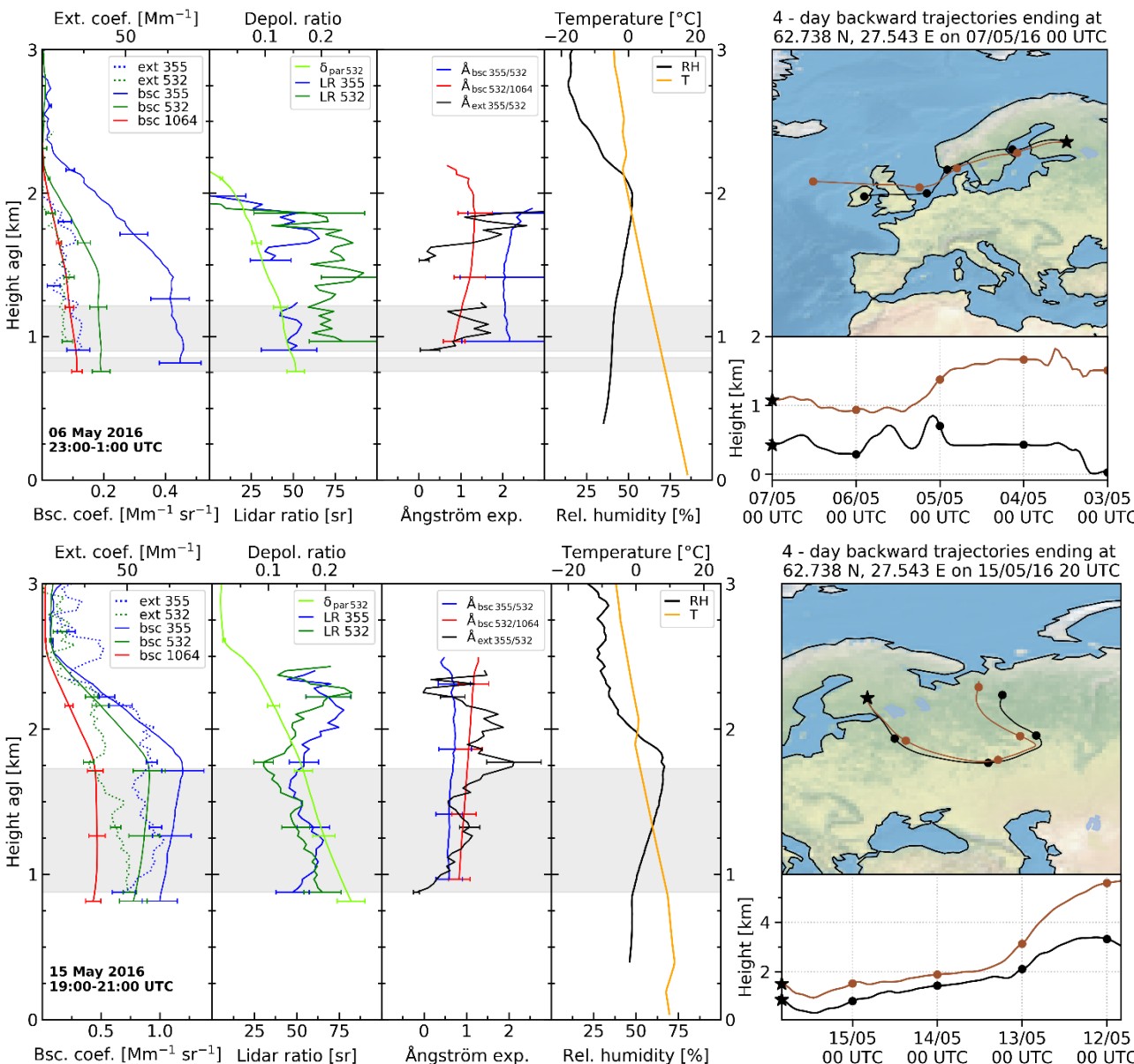

**Figure 3: Two case studies of different pollen mixture cases. Top: Period 1, 06.05.2016 23:00-01:00, only birch pollen was collected by the Hirst-type volumetric air sampler. Bottom: Period 2, 15.05.2016 19:00-21:00, birch and spruce pollen were collected. Left: profiles of backscatter- and extinction coefficients, particle depolarization and lidar ratio, Ångström exponents, relative humidity (derived from lidar measurements) and temperature profiles (18 UTC radio sounding). Right: 4-day HYSPLIT backward trajectories. Defined pollen layers are marked in grey.**

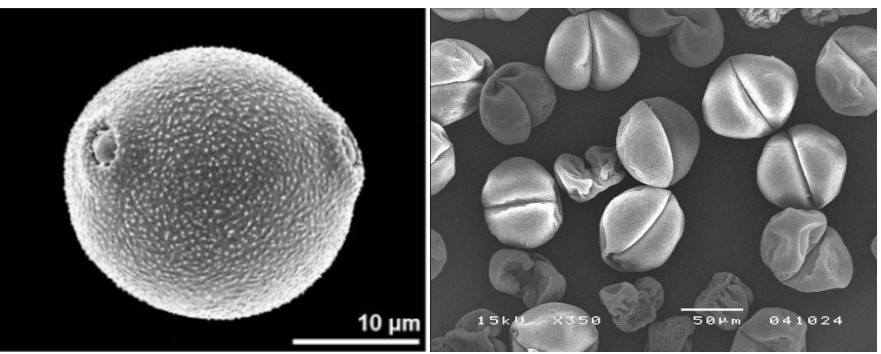

**Figure 4: Pollen micrographs. Left: Birch (Betula pendula) pollen grain (source: Hesse et al., 2009), right: Spruce (Picea abies) pollen grains (source: the Biodiversity Unit of the University of Turku / Kari Kaunisto)**

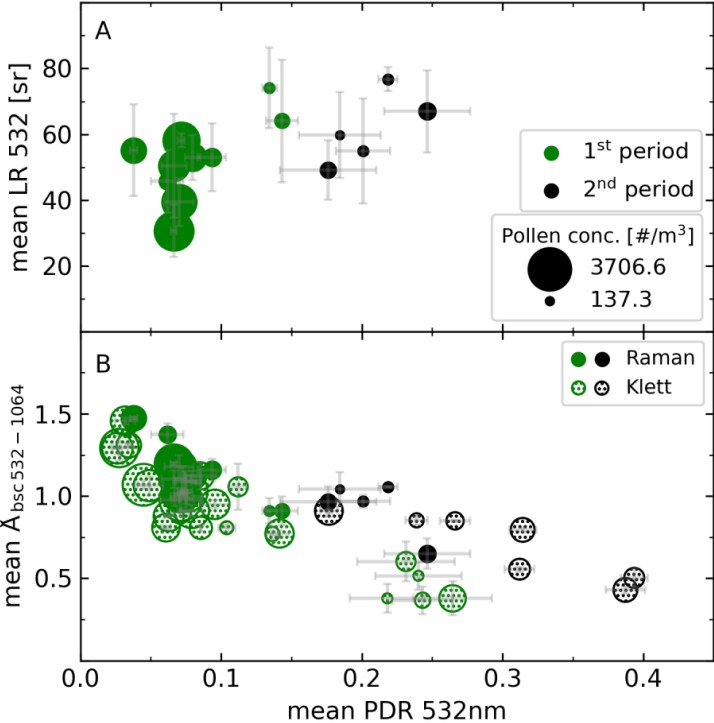

5    **Figure 5: Mean values of optical properties of the detected pollen layers during the period 5 to 15 May, error bars denote the standard deviation, size the pollen concentration. Color denotes the measurement period. Green dots are measured from 5-10 May, black dots from 12-15 May. Solid dots indicate Raman retrievals, dotted markers the Klett solution. (A) Lidar ratio at 532 nm vs particle depolarization ratio at 532 nm, (B) Backscatter-related Ångström exponent at 532-1064 nm vs particle depolarization ratio at 532 nm.**

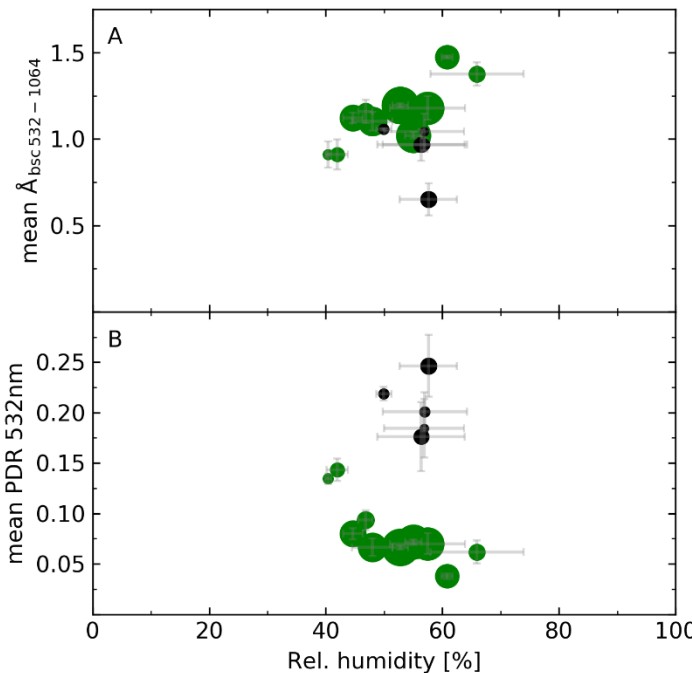

**Figure 6: Dependence of the backscatter-related Ångström exponent at 532-1064 nm (A) and the particle depolarization ratio at 532 nm (B) on the relative humidity for all Raman measurements of pure and mixed birch cases during the first period (5-10 May, green) and the second period (12-15 May, black).**

5  **Table 1: Mean values, range and median of optical properties of the detected pollen layers in the first measurement period: 5-10 May, intense birch pollination period.**

| Parameter | | Mean ± STD | Range | Median |
|---|---|---|---|---|
| **Layer top height [km]** | | 1.3 ± 0.3 | 1.0 - 2.2 | 1.2 |
| **Backscatter coeff.** | 355 nm | 0.7 ± 0.5 | 0.1 - 2.4 | 0.7 |
| **[Mm⁻¹sr⁻¹]** | 532 nm | 0.3 ± 0.2 | 0.1 - 1.0 | 0.3 |
| | 1064 nm | 0.2 ± 0.1 | 0.1 - 0.4 | 0.2 |
| **Extinction coeff.** | 355 nm | 33.0 ± 13.3 | 20.0 - 68.2 | 30.9 |
| **[Mm⁻¹]** | 532 nm | 19.0 ± 6.5 | 11.0 - 34.4 | 19.1 |
| **Lidar ratio [sr]** | 355 nm | 46 ± 8 | 34 - 60 | 46 |
| | 532 nm | 52 ± 12 | 31 - 74 | 53 |
| **PDR** | 532 nm | 0.10 ± 0.06 | 0.03 - 0.26 | 0.08 |
| **Number of pollen layers:** | | all: 41 | Raman: 10 | |

**Table 2: Mean values, range and median of optical properties of the detected pollen layers in the second measurement period: 12-15 May, spruce contaminated period.**

| Parameter | | Mean ± STD | Range | Median |
|---|---|---|---|---|
| **Layer top height [km]** | | 1.3 ± 0.4 | 1 – 2.2 | 1.1 |
| **Backscatter coeff.** | 355 nm | 0.7 ± 0.2 | 0.3 – 1.1 | 0.6 |
| **[Mm$^{-1}$sr$^{-1}$]** | 532 nm | 0.5 ± 0.2 | 0.3 – 0.8 | 0.4 |
| | 1064 nm | 0.3 ± 0.1 | 0.2 – 0.4 | 0.2 |
| **Extinction coeff.** | 355 nm | 52.9 ± 13.1 | 26.9 – 60.9 | 58.5 |
| **[Mm$^{-1}$]** | 532 nm | 40.0 ± 9.5 | 24.6 – 54.6 | 40.2 |
| **Lidar ratio [sr]** | 355 nm | 60 ± 3 | 55 – 64 | 59 |
| | 532 nm | 62 ± 10 | 49 – 77 | 60 |
| **PDR** | 532 nm | 0.26 ± 0.07 | 0.18 – 0.39 | 0.24 |
| **Number of pollen layers:** | | all: 12 | Raman: 5 | |