# Peer review of "Detection and characterization of birch pollen in the atmosphere using multi-wavelength Raman polarization lidar and Hirst-type pollen sampler in Finland"

_Atmospheric Chemistry and Physics, 2019_

## Referee Comment (RC1) · Anonymous Referee #1 · 4 Aug 2019

The paper reports detection and characterization of birch pollen using multi-wavelength Raman lidar.This paper is considered as the first paper detecting pollen using Raman lidar and also multi-wavelength data. The dataset presented is interesting and surely deserves publication. However, I am afraid the paper cannot be published as it is, as a number of points must be clarified. A detailed review follows:

Major Comments 1. Please add the comments about background depolarization ratio at the observation site. The authors mentioned about the effect of dust and biomass particles. And there is no effect by dust and/or biomass particles during study period by satellite data etc. But, there are any explanation about background data except

pollen period. Please add depolarization ratio value in normal day (No pollen, dust and biomass particle).

2. Authors separated the layer as shown in Fig. 2. Only the lowest layer (black color) is considered in the study as mentioned in section 3. But, diurnal variation and maximum heights are different between two pollen period in Fig. 1 (c). In addition, the authors explained that the number of pollen grains is higher in the first period (5-9 May), but the degree of polarization extinction is higher in the second periods, which is explained by the high non-sphericity of spruce pollen. However, only explained reason in the paper is not sufficient. Therefore, it seems that data for other weather elements such as atmospheric boundary layer changes in each period should be added.

Technical comments; 1. Page 3 line30: 10 l/min, change it as 10 L/min. 2. Page 5 line 9 : fro m -> from 3. Page 8 line 9 : please add wavelength for Angstrom exponent. 4. Figure 1 c. Is the scale is log scale for volume depolarization? If yes, could you change it as normal scale? The value of volume depolarization ratio is look too different between two period.

---

## Referee Comment (RC2) · Anonymous Referee #2 · 9 Aug 2019

General comments: The authors presented lidar detection and classification of pollen species (birch and spruce) mainly based on depolarization ratio of them. The data together with backscatter coefficient, extinction coefficient, lidar ratio and Angstrom exponent of pollen that they measured are very basic and useful parameters for lidar studies. However, more focused discussion on the local aerosol except for pollen is required for the paper to be accepted. Comments are given below.

Specific comments: 1 Title: "multi-wavelenth Raman lidar" should be better to be changed to (for example) "multi-wavelength scattering lidar with several lidar parameters", because Raman data was only used to derive water vapor mixing ratio (is this

[Figure]

correct?) that is not main point of the content.

2 Introduction: Please survey and show other methods/techniques of pollen detection such as CCD detection with imaging analysis, fluorescence lidars and others, so that authors can appeal features of their lidar to readers who are not only lidar researchers but also the wide range of them.

3 Two cases: I want to know the reason why you selected only the two cases, the first period on 6 May 2016 between 23:00 and 01:00 and the second period on 15 May 2016 between 19:00 and 21:00. During 5-9 May, most of observations showed much higher concentration of birch pollen than that of first period you selected. And also, it seems better to discuss the influence of spruce pollen by using data in the morning on 13 May (13/05/16, Fig. 1) as spruce pollen concentration was higher than that in the second period.

4 Aerosols: There are several descriptions for the possible existence of local aerosols other than pollen, p. 4 line 13, p. 5 line 22, p. 8, line17 and others. Nevertheless,the content of the paper seems to insist too much on the influence of pollen alone.

5 Hirst-type volumetric air sampler: Relating to #3, please describe the detailed results about solid particles, except for pollens, found/collected with the air sampler. There may be much amount of particles generated from the ground around the local area where the lidar measurements were done. Couldn't you find any particles which showed depolarizationïij§ I think these particles are more important to understand the lidar results than airmass derived with backward trajectories.

6 Range: The line 4 in page 5 describes "we only consider the lowest layer in the following analysis". It was at around 1 km. But Fig. 3 showing range up to 2.5 -3 km makes readers confused. Authors also described that data can be retrieved down to around 500 m. That data is important for comparison with the sampler data.

Technical comments: Technical word and its prefix, such as particle depolarization ratio

and PDR, lidar ratio and PDR, frequently appears at random to each other, please unify them through the whole paper.

---

## Short Comment (SC1) · 10 Sep 2019

Methodology:

-If you speak about concentration of Pollen you refer to the number concentration, but what does this means in terms of the surface concentration and the volume concentration which are the important parameter for backscatter and extinction of light ? Can you clarify with typical sizes from the detected Pollen?

-Furthermore, can you provide a table with the optical and microphysical properties of the different Pollen types you analyse as detected in the laboratory, e.g. the depolar-

ization ratio (Cao et al.), average density of Pollen, refractive index, and fall velocity (would be interesting as most Pollen have good flight behaviour.)

-A discussion on the flight behaviour would be beneficial to relate the lidar measurements to the surface measurements. One could imagine that if particles are to large and to heavy and have thus a high fall velocity they would not make it up to the heights of the lidar observations (this could, e.g. another reason for your changing particle depolarization with height: Is the size distribution of the observed Pollen types nearly one size or can they change?).

-You state that during your observations the lidar ratio of Pollen stays the same while depolarization changes. What theoretical optical properties do you expect from, e.g., scattering calculations? Is there any literature available? Can you explain how the depolarization of the laser light is working at this size range?

Case studies:

-Line 15ff: you state that birch Pollen (20-30 mum) are smaller than spruce (70-100 mum) and therefore the Ångström exponent is higher when only birch is prevailing. However, considering the laser wavelength and the size of the particles, I would expect an Ångström exponent of 0 for all particles (size parameter well above on). Can you state on this?

-Derived optical parameters: Are the Pollen probably aligned in a certain way that they can fly better and might therefore a difference in the observed optical properties in comparison to laboratory measurements occur?

Idea:

-As you have determined the particle depolarization ratio of pure birch and you know the contribution of both constituents during the mixed birch/spruce period, couldn't you calculate the theoretical depolarization ratio of pure spruce by applying the depolarization separation formula of Tesche, 2009, JGR . If I am right, you will need to convert

your number concentrations to surface concentrations to obtain the backscatter fraction.

---

## Author Comment (AC1) · 10 Sep 2019

**Letter of Reply to Referee 1**

Thank you for carefully reading the manuscript and providing useful suggestions to improve the paper.
The replies to your comments are given below.

**Major Comments 1. Please add the comments about background depolarization ratio at the observation site. The authors mentioned about the effect of dust and biomass particles. And there is no effect by dust and/or biomass particles during study period by satellite data etc. But, there are any explanation about background data except pollen period. Please add depolarization ratio value in normal day (No pollen, dust and biomass particle).**

To check the background aerosol at our measurement site, lidar measurements during different times of the year have been checked. The particle depolarization ratio of the ground-near aerosol layer during the winter months of 2015 and 2016 are shown in Fig.1a. The winter time was selected to ensure the absence of pollen. Backward trajectories have been checked and measurements with distinct aerosol layers were omitted. The mean PDR during normal days without influence of pollen or other depolarizing aerosols is 4±1%, which is smaller than during the pollen-influenced period. Also pollen-free periods during spring and summer time, which means no pollen have been counted by the Hirst-type pollen collector, have been checked and are shown in Fig1 b, c. The mean values of PDR during those periods was 3.5±1% in March/May and 3.9±1% in July/August. However, during spring and summer the absence of pollen in the atmosphere cannot be confirmed with absolute certainty even if there are no pollen collected by the Hirst-type sampler.

As the depolarization ratio of cases without detected pollen on ground-level or other depolarizing aerosol is below 4% and therefore significantly lower than during the presented pollination period, the particle depolarization ratio can be used to detect the presence of pollen in the atmosphere.
Information about background aerosol was added in section 4.2., as:

"*However the effect of the background particles has to be considered. Lidar measurements during the winter months of 2015 and 2016 and during pollen-free periods in spring and summer 2016 have been analyzed to determine the effect of background aerosol at our measurement site. During winter time the absence of pollen can be ensured, but there is a possibility that pollen also have been present in the atmosphere during spring and summer when no pollen were detected by the Hirst-type sampler on ground. Nevertheless, values of mean PDR at 532 nm are below 4% during all analyzed periods with no observed pollen concentration. Since the PDR during the pollination period is significantly higher than the PDR of the background aerosol, the depolarization ratio can be used as an indicator for detecting the presence of pollen.*"

[Figure]

*Figure 1 Histograms of mean PDR within the lowest aerosol layer during January 2015 and 2016 (A) and pollen-free cases during March/May 2016 (B) and July/August 2016 (C)*

**2. Authors separated the layer as shown in Fig. 2. Only the lowest layer (black color) is considered in the study as mentioned in section 3. But, diurnal variation and maximum heights are different between two pollen period in Fig. 1 (c). In addition, the authors explained that the number of pollen grains is higher in the first period (5-9 May), but the degree of polarization extinction is higher in the second periods, which is explained by the high non-sphericity of spruce pollen. However, only explained reason in the paper is not sufficient. Therefore, it seems that data for other weather elements such as atmospheric boundary layer changes in each period should be added.**

We checked the meteorological variation during our campaign. Mean values of the diurnal cycle of temperature and relative humidity at our measurement site 2 m above ground, during the first (green) and second (black) period are shown in Fig.2. Green and grey shaded areas indicate the variability during the two periods. The higher pollen concentration during the first period is likely related to the higher temperature and lower relative humidity during this period. In previous studies a correlation between pollen concentration and temperature and relative humidity has been shown. The different mixture of pollen types can be explained by the changing wind direction during the periods (see Fig.3). The most frequent wind direction in this first period was northwest. In the second period the air masses were mainly advected from southeast. The boundary layer height only changes slightly during the selected period and there are no significant differences between the two periods. A paragraph about the meteorological conditions during both periods is added in section 3, as:

*"Those variations can be explained by the different meteorological conditions during the two periods. A big difference of the predominant wind direction in the two periods has been observed, which may cause the different mixture of pollen types. The most frequent wind direction in period 1 was northwest, whereas in period 2 the air masses were mainly advected from southeast. When comparing the diurnal cycle of temperature and relative humidity measured at ground level, we found higher temperature values and lower relative humidity in period 1 than period 2. Temperature and pollen concentration have been shown to be positive correlated whereas pollen concentration and relative humidity show a negative correlation (Bartková-Ščevková, 2003). The different pollen concentration could therefore be explained by variations of temperature and humidity."*

[Figure]

*Figure 2 Diurnal cycle of mean temperature (top) and relative humidity (bottom) during the first period (green) and second period (black). The standard deviation is shown by the green and grey shaded area.*

*Figure 3 Wind direction during the first period (left) and second period (right).*

**Technical comments; 1. Page 3 line30: 10 l/min, change it as 10 L/min. 2. Page 5 line 9: fro m -> from 3. Page 8 line 9: please add wavelength for Angstrom exponent. 4. Figure 1 c. Is the scale is log scale for volume depolarization? If yes, could you change it as normal scale? The value of volume depolarization ratio is look too different between two periods.**

Comments 1-3 are applied to the manuscript. We prefer to keep Fig. 1c in log scale, because the depolarization ratio, especially in the first period, is better visible as can be seen in the comparison between linear and log scale shown in Fig.4. However, we changed the color bar label of Fig.1c in the manuscript to make the log scale better recognizable.

[Figure]

*Figure 4 Comparison of Figure 1 without logarithmic scale (left) and with logarithmic scale (right).*

---

## Author Comment (AC2) · 10 Sep 2019

**Letter of Reply to Referee 2**

Thank you for carefully reading the manuscript and providing useful suggestions to improve the paper.
Replies to your comments are given below.

**General comments: The authors presented lidar detection and classification of pollen species (birch and spruce) mainly based on depolarization ratio of them. The data together with backscatter coefficient, extinction coefficient, lidar ratio and Angstrom exponent of pollen that they measured are very basic and useful parameters for lidar studies. However, more focused discussion on the local aerosol except for pollen is required for the paper to be accepted. Comments are given below.**

**Specific comments:**

**1 Title: "multi-wavelength Raman lidar" should be better to be changed to (for example) "multi-wavelength scattering lidar with several lidar parameters", because Raman data was only used to derive water vapor mixing ratio (is this correct?) that is not main point of the content.**

The used lidar instrument is a multi-wavelength Raman polarization lidar. In addition to the determination of the water vapor mixing ratio, Raman measurements are also used to derive the lidar ratio, which is an important property for particle typing. We therefore prefer to keep the specification of our lidar system in the title. However the title was changed to "Detection and characterization of birch pollen in the atmosphere using multi-wavelength Raman polarization lidar and Hirst-type pollen sampler in Finland" to emphasis the synthesis of lidar and pollen collector.

**2 Introduction: Please survey and show other methods/techniques of pollen detection such as CCD detection with imaging analysis, fluorescence lidars and others, so that authors can appeal features of their lidar to readers who are not only lidar researchers but also the wide range of them.**

A paragraph about other pollen detection methods was added in the introduction, as:

*"The majority of these stations operate Hirst-type volumetric air sampler. These traditional pollen traps are operated manually, which requires human resources and is time consuming. In the recent years, novel techniques have been developed to enable automated pollen monitoring and reduce workload. Those techniques use, for example, automated image recognition (Oteros et al., 2015) or fluorescence spectra (Crouzy et al., 2016) to identify pollen types and could enable a systematic pollen monitoring on ground level in near real-time."*

**3 Two cases: I want to know the reason why you selected only the two cases, the first period on 6 May 2016 between 23:00 and 01:00 and the second period on 15 May 2016 between 19:00 and 21:00. During 5-9 May, most of observations showed much higher concentration of birch pollen than that of first period you selected. And also, it seems better to discuss the influence of spruce pollen by using data in the morning on 13 May (13/05/16, Fig. 1) as spruce pollen concentration was higher than that in the second period.**

The first case was chosen because we assume less contamination with anthropogenic aerosol during 6 May than during the period 8-9 May, when the concentration of birch pollen was much higher. During those days trajectories show that air masses travelled over the European continent e.g. Germany, which could have caused a contamination with anthropogenic aerosols. We want to show a case with minimal contamination of other aerosols. The second case study was chosen because we want to show Raman measurements (which are only possible during night time with less background light) with all lidar-derived parameters. 15 May was the only possible period suitable for lidar analysis (i.e. no low clouds and sufficient signal-to-noise ratio) although the spruce pollen concentration is higher on 13 May.

**4 Aerosols: There are several descriptions for the possible existence of local aerosols other than pollen, p. 4 line 13, p. 5 line 22, p. 8, line17 and others. Nevertheless, the content of the paper seems to insist too much on the influence of pollen alone.**

To check the background aerosol at our measurement site, lidar measurements during different times of the year have been checked. The particle depolarization ratio of the ground-near aerosol layer during the winter months of 2015 and 2016 are shown in Fig.1a. The winter time was selected to ensure the absence of pollen. Backward trajectories have been checked and measurements with distinct aerosol layers were omitted. The mean PDR during normal days without influence of pollen or other depolarizing aerosols is 4±1%, which is smaller than during the pollen-influenced period. Also pollen-free periods during spring and summer time, which means no pollen have been counted by the Hirst-type pollen collector, have been checked and are shown in Fig1 b, c. The mean values of PDR during those periods was 3.5±1% in March/May and 3.9±1% in July/August. However, during spring and summer the absence of pollen in the atmosphere cannot be confirmed with absolute certainty even if there are no pollen collected by the Hirst-type sampler.

As the depolarization ratio of cases without detected pollen on ground-level or other depolarizing aerosol is below 4% and therefore significantly lower than during the presented pollination period, the particle depolarization ratio can be used to detect the presence of pollen in the atmosphere.

Information about background aerosol was added in section 4.2., as:

"*However the effect of the background particles has to be considered. Lidar measurements during the winter months of 2015 and 2016 and during pollen-free periods in spring and summer 2016 have been analyzed to determine the effect of background aerosol at our measurement site. During winter time the absence of pollen can be ensured, but there is a possibility that pollen also have been present in the atmosphere during spring and summer when no pollen were detected by the Hirst-type sampler on ground. Nevertheless, values of mean PDR at 532 nm are below 4% during all analyzed periods with no observed pollen concentration. Since the PDR during the pollination period is significantly higher than the PDR of the background aerosol, the depolarization ratio can be used as an indicator for detecting the presence of pollen.*"

[Figure]

Figure 1 Histograms of mean PDR within the lowest aerosol layer during January 2015 and 2016 (A) and pollen-free cases during March/May 2016 (B) and July/August 2016 (C)

**5 Hirst-type volumetric air sampler: Relating to #3, please describe the detailed results about solid particles, except for pollens, found/collected with the air sampler. There may be much amount of particles generated from the ground around the local area where the lidar measurements were done. Couldn't you find any particles which showed depolarization? I think these particles are more important to understand the lidar results than airmass derived with backward trajectories.**

The analysis of the collection tape of the Hirst-type air sampler only involves the examination of pollen grains. Other particles are not counted or analyzed. During the selected pollination period no other instruments measuring the depolarization were available. It is therefore not possible to make any statements about other depolarization particles close to the ground. However, the effect of depolarizing particles originating from the ground, can be considered negligible at the latitudes the lidar data is used. Also if such depolarizing particles were frequently present, this would be visible in the PDR values of days without pollen.

**6 Range: The line 4 in page 5 describes "we only consider the lowest layer in the following analysis". It was at around 1 km. But Fig. 3 showing range up to 2.5 -3 km makes readers confused. Authors also described that data can be retrieved down to around 500 m. That data is important for comparison with the sampler data.**

The lowest layer is used because the highest amount of local pollen is likely located in the layer closest to the ground. However our profiles cannot be extended down to ground-level. Theoretically profiles can be determined with overlap correction down to 500 m. However in this study the lower limit of our profiles is at around 800 m with a vertical smoothing of 25 bins (750 m) and a time average of 2 hours. Data below could not be used for comparison with the sampler data. The profile ranges used for calculation of mean values (pollen layer) is marked grey in Fig.3. Nevertheless the profiles are shown up to 3 km to give more information about the vertical aerosol distribution. A comment about the pollen layers shown in grey was added to the caption of Fig.3 and clarification that profiles are only reliable starting from 800 m in this study was added in the section 2.1, as:

*"In this study, the lower limit of reliable profiles of vertically smoothed and temporally averaged optical properties is at around 800 m."*

**Technical comments: Technical word and its prefix, such as particle depolarization ratio and PDR, lidar ratio and LR, frequently appears at random to each other, please unify them through the whole paper.**

Technical words and their abbreviations have been unified.

---

## Author Comment (AC3) · 14 Oct 2019

**Reply to Dr. Holger Baars**

Thank you for your comments! Below you find our detailed response to each question.

**Methodology:**

**1) If you speak about concentration of Pollen you refer to the number concentration, but what does this means in terms of the surface concentration and the volume concentration which are the important parameter for backscatter and extinction of light? Can you clarify with typical sizes from the detected Pollen?**

Thank you for this comment! Typical sizes of pollen are given in the results section. Also a comment about the volume concentration was added when explaining the effect of spruce pollen on the measured values despite the low number concentration contribution, as:

*The Ångström exponent in the second period is around 0.8, whereas it is around 1 in the first period, demonstrating the effect of the larger spruce pollen (~90-110 μm), even with a small contribution (~14%) to the total pollen number concentration. But considering the different volume of birch and spruce pollen grains, the contribution of spruce to the total volume concentration exceeds 75% in the second period, which explains the large effect of spruce pollen on the measured optical properties even with a small number concentration.*

**2) Furthermore, can you provide a table with the optical and microphysical properties of the different Pollen types you analyse as detected in the laboratory, e.g. the depolarization ratio (Cao et al.), average density of Pollen, refractive index, and fall velocity (would be interesting as most Pollen have good flight behaviour.)**

Thank you, this is a good suggestion, but optical and microphysical properties of both of the pollen types are either not reported in literature yet or vary significantly as for example the hydration of pollen grains varies and affects the density and therefore the fall velocity. A comment about the fall velocity of birch pollen was added in the result section, but because spruce pollen are much less studied and to our knowledge there are no laboratory measurements of optical properties of spruce pollen yet, we prefer not to provide a table as it would be incomplete and it is not crucial for the understanding of our study.

**3) A discussion on the flight behaviour would be beneficial to relate the lidar measurements to the surface measurements. One could imagine that if particles are too large and too heavy and have thus a high fall velocity they would not make it up to the heights of the lidar observations (this could, e.g. another reason for your changing particle depolarization with height: Is the size distribution of the observed Pollen types nearly one size or can they change?).**
Airborne pollen are designed in a way so that the grains can be dispersed by wind. Bigger and thus heavier pollen grains possess air bladders, which increase the surface area of the grain without adding much mass. This increases the amount of drag and decrease the settling velocity of the pollen grains. In this way it is also possible for big pollen grains to be lifted and dispersed by wind. In various studies e.g. Skjøth, 2007; Rousseau, 2008; Szczepanek, 2017, the long range transport of pollen has been shown. The size distribution of pollen within one pollen type is usually narrow. The size varies only little within one specie (Mäkelä, 1996; Kishchenko and Tikhova, 2019). A comment about the flight behavior of pollen is added in section 4.1., as:

*Pollen are low density particles, which makes them more sensitive to air currents, reduces the settling velocity and allows them to be lifted by turbulent air flows. Birch pollen, for example, have a gravitational settling velocity of around 1 cm s⁻¹ (Sofiev et al., 2006). This settling velocity is similar to anthropogenic aerosol smaller than 10 μm (PM10) although birch pollen grains are more than twice the size. The air bladders on the bigger spruce pollen grains increase the surface area of the grain without adding much mass, and therefore decrease the settling velocity. Hence, even those big pollen grains can be lifted up to several kilometers and be dispersed by wind over thousands of kilometers as have been shown by several studies on the long distance transport of pollen (Rousseau et al., 2008; Skjøth et al., 2007; Szczepanek et al., 2017).*

**4) You state that during your observations the lidar ratio of Pollen stays the same while depolarization changes. What theoretical optical properties do you expect from, e.g., scattering calculations? Is there any literature available? Can you explain how the depolarization of the laser light is working at this size range?**

Unfortunately there are no theoretical optical properties for the observed pollen types under ambient conditions, e.g. hydrated pollen. The paper by Cao et al. provides experimental values for various pollen types but doesn't have any values for example for spruce pollen. Furthermore dry pollen grains were used which have different optical properties. We will try to obtain scattering simulations for future studies as theoretical reference values for the optical properties of different pollen types under different environmental conditions.

**Case studies:**

**5) Line 15ff: you state that birch Pollen (20-30 mum) are smaller than spruce (70-100mum) and therefore the Ångström exponent is higher when only birch is prevailing. However, considering the laser wavelength and the size of the particles, I would expect an Ångström exponent of 0 for all particles (size parameter well above on). Can you state on this?**

The Ångström exponent we are measuring is not the Ångström exponent of pure pollen. Also background aerosol contributes to the observed optical properties. We assume that pollen have a contribution to the Angström, but the derivation from 0 is caused by the background aerosol. We are working on separation techniques of pollen and background aerosol to gain pure pollen properties without the effect of background aerosol, but this is not in the scope of this study.

**6) Derived optical parameters: Are the Pollen probably aligned in a certain way that they can fly better and might therefore a difference in the observed optical properties in comparison to laboratory measurements occur?**

Pollen grains with air bladder, e.g. spruce pollen, align with their air bladders upwards when drifting in the air. (Schwendemann, 2007) and also the almost spherical birch pollen were found to have an orientation when falling (Sassen, 2011; Tränke and Mielke, 1994). This could also partly explain differences between ambient measurements and laboratory experiments, this thought is added in the manuscript as:

*Also the orientation of the pollen grains in the atmosphere has to be considered. Pollen with air bladders, e.g. spruce pollen, are known to align with their air bladders upwards when drifting in the air (Schwendemann et al., 2007) and also a certain orientation of the almost spherical birch pollen grains was observed (Sassen, 2011; Tränkle and Mielke, 1994). This could cause differences in the measured optical properties if the orientation of*

*the particles in laboratory experiments is not considered and the irregularly shaped particles are observed from different angles.*

 **Idea:**

**7) As you have determined the particle depolarization ratio of pure birch and you know the contribution of both constituents during the mixed birch/spruce period, couldn't you calculate the theoretical depolarization ratio of pure spruce by applying the depolarization separation formula of Tesche, 2009, JGR. If I am right, you will need to convert your number concentrations to surface concentrations to obtain the backscatter fraction.**

This is a good idea, but we don't determine the particle depolarization of pure birch. We are providing the depolarization ratio for a period with birch pollen including background aerosol. The fraction/contribution of background aerosol cannot be determined with the setup during the presented campaign, which complicates the determination of the pure spruce depolarization. However, we are working on a method to estimate the pure PDR of different pollen and it will be the topic of a future paper which is going to be submitted soon.

---

## Author Comment (AC4) · 14 Oct 2019

**Reply to the additional comments of referee 2**

**Comments to the reply:**

**#2 Introduction**
**Please consider referring the following two papers on pollen monitoring with a fluorescence lidar, S.C. Richardson et al, Science of the Total Environment 696 (2019) 133906 and Y. Saito et al, Remote Sensing 10 (2018) 1533. These will help clarify the characteristics of your method/system.**

Thank you. The mentioned paper references were added to clarify the use of fluorescence lidars to detect pollen.

**#3 Two cases**
**I think it would be better to add such a description you replied. Readers wonder why only two short cases (2hours x 2days) were focused on in the study. You have made a great effort on the long time observation for 11 days.**

An explanation was added, as:

*In the choice of case studies, backward trajectories have been considered to select cases with minimal contamination with of other aerosol. Furthermore nighttime Raman measurements were chosen to present all lidar-derived parameters including the retrieved LR profile.*

**#5 Hirst-type volumetric air sampler**
**Is "the effect of depolarizing particles originating from the ground, can be considered negligible at the latitude the lidar data is used." correct? Various kinds of ground particles can be carried up to several kilometers where lidars can work well. I think some ones of them show a certain depolarizing. Does "Also if such depolarizing particles were frequently present, this would be visible in the PDR values of days without pollen." mean that your lidar only reacts to especially pollen? It is hard to think so.**

We meant that if we would have a lot of depolarizing particles in the height of our observation, we would observe high depolarization ratios also during periods without pollen, because other particles sources from the ground are not linked to the pollination time. Measurements without pollen however show a relatively low PDR of around 4%.